

# The regulatory roles and mechanisms of the transcription factor FOXF2 in human diseases

Qiong Wu, Wei Li and Chongge You

Laboratory Medicine Center, Lanzhou University Second Hospital, Lanzhou, China

## ABSTRACT

Many studies have focused on the relationship between transcription factors and a variety of common pathological conditions, such as diabetes, stroke, and cancer. It has been found that abnormal transcription factor regulation can lead to aberrant expression of downstream genes, which contributes to the occurrence and development of many diseases. The forkhead box (FOX) transcription factor family is encoded by the *FOX* gene, which mediates gene transcription and follow-up functions during physiological and pathological processes. FOXF2, a member of the FOX transcription family, is expressed in various organs and tissues while maintaining their normal structural and functional development during the embryonic and adult stages. Multiple regulatory pathways that regulate FOXF2 may also be controlled by FOXF2. Abnormal FOXF2 expression induced by uncontrollable regulatory signals mediate the progression of human diseases by interfering with the cell cycle, proliferation, differentiation, invasion, and metastasis. FOXF2 manipulates downstream pathways and targets as both a pro-oncogenic and anti-oncogenic factor across different types of cancer, suggesting it may be a new potential clinical marker or therapeutic target for cancer. However, FOXF2's biological functions and specific roles in cancer development remain unclear. In this study, we provide an overview of FOXF2's structure, function, and regulatory mechanisms in the physiological and pathological conditions of human body. We also discussed the possible reasons why FOXF2 performs the opposite function in the same types of cancer.

# INTRODUCTION

The forkhead box (FOX) transcriptional factor family is characterized by a conserved DNA binding domain with a winged helix structure. They serve as key regulators in various biological and pathological processes, such as embryogenesis, cell cycle, signal transduction, maintenance of differentiated cell states, DNA damage responses, and tumorigenesis (*Katoh & Katoh, 2004*). There are 19 FOX transcription factor subfamilies, starting with FOXA and ending with FOXS (*Huang et al., 2020*). Each member plays a different role in the normal development and pathological changes of various systems of the human body. FOXA1, which helps establish and maintain cellular identity, is overexpressed in lung and esophageal cancers (*Lin et al., 2002*). FOXF1 is required for gut, liver, and pulmonary growth during

Corresponding author
Chongge You, youchg@lzu.edu.cn

the early stages of embryo development (*Aitola et al., 2000*). *FOXF1* mutation is associated with alveolar capillary dysplasia and Hirschsprung's disease (*Goel et al., 2016*; *Stankiewicz et al., 2009*). FOXM1, a pivotal cell cycle regulator, is overexpressed due to Sonic Hedgehog (Shh) signaling activation in basal-like breast cancer (BLBC), diffuse large B-cell lymphoma (DLBCL), and pancreatic cancer (*Golson & Kaestner, 2016*). In chondrocytes, FOXO can affect normal skeletal development and immune response (*Cabrera-Ortega , 2017*; *Kim et al., 2018*).

FOXF2, a member of the FOXF subfamily, has similar functions to FOXF1, such as regulating embryonic development and disease occurrence (*Aitola et al., 2000*). Previous studies on FOXF2 found that mice lacking FOXF2 with cleft palates died perinatally because their oral cavity could not form negative pressure, leading to difficulties in sucking. Filling the gastrointestinal tract with gases and liquids caused over-pressurization, which compressed the organs and cut off the blood supply (*Ormestad et al., 2006*). Additionally, it has been found that FOXF2 is abnormally expressed in other pathological conditions.

Cancer is a genetic or epigenetic disease that alters cell functions and metabolic activities, and the main cause of death in cancer patients is metastasis (*Gupta & Massagué, 2006*). The epithelial-mesenchymal transition (EMT) performed by EMT transcription factors (EMT-TFs) is abnormally activated in cancer and increases cell metastasis (*Stemmler et al., 2019*). FOXF2 mediates cancer progression, either as a tumor-promoter or a tumor-suppressor, by regulating EMT, cell cycles, and cell matrix production (*Aitola et al., 2000*). Abnormal FOXF2 expression causes uncontrolled regulation of downstream genes and associated signaling pathways, including the Wnt/$\beta$-catenin and transforming growth factor-$\beta$ (TGF$\beta$)-SMAD pathways. This can lead to deviations in cell proliferation, differentiation, and metastasis, which can catalyze cancer occurrence and development (*Higashimori et al., 2018*). Some epigenetic modifications can affect pathological processes by regulating transcription factor expression (*Hirata et al., 2013*). Different FOXF2 functions and expression forms have warning effects during the tumor stage, prognosis, and treatment of cancer.

## SURVEY METHODOLOGY

All manuscripts cited in this article came from professional databases including PubMed, Web of Science, and Google Scholar. The inclusion criteria required that the article had been peer-reviewed and looked at the relationship between abnormal FOXF2 expression and cancer. Our keywords when searching across related articles included "FOXF2 and diseases," "FOXF2 and cancer," "FOXF2 transcription factor," "FOXF2 and clinical prognosis," and" epigenetic modification regulation". We considered and critically analyzed both basic research and clinical research articles in order to provide a comprehensive and unbiased selection of the literature.

### Basic overview of FOXF2

FOXF2 is a 444 amino-acid protein encoded by the *FOXF2* gene, located on chromosome 6p25. FOXF2 regulates cell growth, differentiation, and metastasis by binding to DNA in the nucleus through its forkhead domain, consisting of about 100 amino acids and two

independent C-terminal activation domains (*Myatt & Lam, 2007*). FOXF2 is specifically expressed in the mesenchyme, which is adjacent to the epithelium in organs derived from the splanchnic mesoderm and plays a critical role in embryo and organ development. FOXF2 maintains cellular homeostasis by regulating EMT, cell cycles, and cell matrix production (*Aitola et al., 2000*).

In the central nervous system (CNS), FOXF2 is specifically expressed in CNS pericytes, which can activate genes related to the blood–brain barrier (BBB) and promote BBB formation. Additionally, FOXF2 regulates the interaction between pericytes and endothelial cells and the production of extracellular matrix (ECM) in the basement membrane of blood vessels through platelet-derived growth factor (PDGF)/platelet-derived growth factor receptor$\beta$ (PDGFR$\beta$) and TGF$\beta$-SMAD2/3 signaling pathways (*Gaengel et al., 2009*; *Leveen et al., 1994*). Pulmonary surfactant-associated proteins $A/B/C$ (SP$A/B/C$ ) are important lung-specific expression proteins that can be activated by FOXF2 (*Hellqvist et al., 1998*; *Hellqvist et al., 1996*). In lung, FOXF2 is highly expressed in mesenchymal cells that form alveolar vascular endothelium and connective tissue fibroblasts around bronchial epithelium, and cooperates with PDGF and wingless-type MMTV integration site family member 2 (Wnt2) -wingless-type MMTV integration site family member 7b (Wnt7b) signaling to promote the differentiation of mesenchymal cells (*Fulton et al., 2018*). In the respiratory system, FOXF2 is associated with mesenchyme and vasculature development, regulation of cell proliferation, and ECM remodeling (*Fulton et al., 2018*). Furthermore, FOXF2 regulates the normal development of the myenteric plexus and smooth muscle proliferation via PDGF/PDGFR, Wn$t/\beta-c$atenin, and Shh signaling during intestinal development (*Bolte et al., 2015*).

## FOXF2 and non-cancer diseases
### *Role of FOXF2 in craniofacial and special receptor diseases*
During embryogenesis, FOXF2 is expressed in the mesenchymal cells of the palatal shelf and the muscular layer of the tongue (*Aitola et al., 2000*). FOXF2 dysregulation attenuates TGF$\beta$ signaling by reducing phosphorylation of TGF$\beta$2 protein and SMAD2/3, which inhibits ECM (especially collagen) production and cell proliferation, leading to loss of the soft palate, lateral movement of the maxillary skeleton, and failed palatal shelf fusion (*Nik et al., 2016*). Additionally, aberrant FOXF2 expression also leads to small tongue loss, velopharyngeal insufficiency, and abnormal craniofacial development (*Seselgyte et al., 2019*). The Shh- Patched 1 (PTCH 1)/ Patched 2 (PTCH 2) receptor-smoothened (SMO)-glioma-associated oncogene homolog (Gli) signaling pathway interacts with FOXF2 to regulate craniofacial development, and abnormal signaling results in decreased FOXF2 in the stroma that causes craniofacial dysplasia (*Lan & Jiang, 2009*).

FOXF2 also has specific expression in the anterior segment of the eye and the mesenchymal cells of the inner ear. Abnormal FOXF2 expression is associated with Axenfeld-Rieger syndrome, which causes hearing and visual impairment, anomalies of the anterior chamber of the eye and anterior segment with iris defects, and neurocognitive disorders (*Kapoor et al., 2011*). FOXF2 knockout reduces the expression of mouse cochlear-related developmental genes Eya1 and Pax3 (*Bademci et al., 2019*). Single nucleotide

polymorphisms (SNPs) have been shown to affect normal cranial and facial development. *FOXF2* variants rs1711968 and rs732835 were shown to be associated with non-syndromic cleft lip with or without cleft palate in an Asian population (*Bu et al., 2015*). Homozygous *FOXF2* variant c.325A>T (p.I109F) causes sensorineural hearing loss (SNHL) and incomplete partition type I anomaly of the cochlea (*Bademci et al., 2019*).

### Role of FOXF2 in stroke

The major causes of death in China are vascular disease, cancer, and chronic respiratory disease (*Wang et al., 2005*). Developments in medical technology and drugs have decreased both the incidence and mortality of strokes, but they are still a serious health threat (*Guzik & Bushnell, 2017*). Strokes are classified into ischemic and hemorrhagic types, and ischemic stroke is more common. Ischemic stroke is divided into five categories: large-artery atherosclerotic (LAA), cardiogenic cerebral embolism, small arterial occlusion (SAO), other causes, and unknown causes (*Guzik & Bushnell, 2017*). TSPAN2, MMP12, CDC5L, and HDAC9 have been shown to be associated with LAA ischemic stroke, while ALDH2, FOXF2, and PRKCH are associated with SAO ischemic stroke (*Chauhan & Debette, 2016*). Specifically, FOXF2 variant rs12204590 increases SAO ischemic stroke susceptibility in European populations (*Chauhan & Debette, 2016*) and rs1711972 increases LAA ischemic stroke susceptibility in Han Chinese (*Shi et al., 2017*). Furthermore, FOXF2 deletion causes demyelinating diseases of cerebral white matter and nervous lesion (*Kapoor et al., 2011*).

### Role of FOXF2 in gastrointestinal diseases

During intestinal development, FOXF2 activates fibroblasts to produce ECM and maintain intracellular homeostasis (*Ormestad et al., 2006*). In the absence of FOXF2, high levels of PDGF and PDGFR promoted the expansion of myenteric plexus and lateral smooth muscle proliferation (*Bolte et al., 2015*). However, this conclusion was in contrast with a report that found that FOXF2 −/− mice lacked ganglia in their colons (*Ormestad et al., 2006*). This result might have been due to using different mouse models. In summary, aberrant FOXF2 expression causes intestinal growth retardation, matrix destruction (*Ormestad et al., 2006*), length reduction, smooth muscle hyper-proliferation, thickening of the outer longitudinal smooth muscle layer (*Bolte et al., 2015*), attenuation of the innervation of distal nerves and muscles that result in aganglionic megacolon and colorectal muscle hypoplasia, and agangliosis (*Ormestad et al., 2006*) in gastrointestinal diseases. Besides, FOXF2 increases the expression of smooth muscle genes and proteins, such as myocardin, serum response factor (SRF), and contractile proteins Myh11, Acta2, calponin, and SM22$\alpha$. Conversely, decreased FOXF2 expression results decreases the expression of smooth muscle contractile proteins, myocardin and SRF, leading to gastric liquid emptying disorders (*Herring et al., 2019*).

### Role of FOXF2 in osteoporosis

*FOXF2* mRNA can be detected in the mesenchymal cells of developing limbs and backbones during the embryonic period. Specific FOXF2 expression in the differentiated dorsal limb tendon is closely related to limb tendon development (*Liu et al., 2015*). The TNF-$\alpha$, IL-1$\beta$, and IL-6 secretion levels increased and the IL-10 level decreased in particle-induced

osteolysis (PIO) mice when compared with normal mice. FOXF2 expression deficiencies activated NF-KB signaling, leading to inflammation and osteolysis. However, FOXF2 overexpression repressed the inflammatory response and prevented the occurrence of osteoporosis. MiR-130b targets FOXF2 to promote the activation of inflammatory factors (TNF-$\alpha$, IL-1$\beta$, and IL-6), thereby affecting FOXF2/NF-KB signaling pathway (*Zheng, Bu & Wang, 2018*).

## FOXF2 and other diseases

FOXF2 upregulation and SMAD6 downregulation are pathological reasons for excessive ECM deposition, fibrosis formation and uterine cavity deformation in the endometrium of intrauterine adhesion (IUA) patients. Through negative regulation of TGF $\beta$ signaling, SMAD6, a downstream mediator of TGF$\beta$, inhibits EMT, fibrosis, and COL5A2 and COL1A1 expression. FOXF2, also a TGF$\beta$ target, cooperated with TGF$\beta$ signaling to promote ECM formation. FOXF2 downregulation and SMAD6 upregulation affected the phase transition from G0/G1 to S, induced by TGF$\beta$1 in vitro and decreased the number of endometrial fibroids encoded by COL5A2 in vivo (*Chen et al., 2020*).

In adipocytes, FOXF2 plays a significant role in glucose metabolism, but few studies have explored this. Insulin receptor substrate 1 (IRS1) is a key factor in the insulin metabolic pathway, and its polymorphisms have been shown to be associated with type 2 diabetes or other related phenotypes. FOXF2 reduces the glucose intake of adipose tissue by inhibiting phosphorylation on Ser307 and Ser612 of IRS1, which is the downstream target of FOXF2. Therefore, FOXF2 can promote insulin secretion and mediate insulin resistance (*Westergren et al., 2010*).

FOXF2 expressed in the posterior second heart field regulates Txb5, which is responsible for the atrioventricular septum, cardiomyocyte proliferation, cardiac contraction, and rhythm regulation in a Shh-dependent manner. Abnormal FOXF2 expression causes atrioventricular septal defects (AVSDs) in mice (*Hoffmann et al., 2014*).

## FOXF2 and human cancers

FOXF2 exerts different, and sometimes even opposite regulatory effects in different tumor types or subtypes of the same tumor by mediating cellular behavior, including cycles, proliferation, invasion, and metastasis, with tissue and progression specificity. FOXF2 also partially regulates changes in tumor microenvironmental components and their associations with tumor cells. FOXF2 can be used as both a tumor-promoting factor and a tumor suppressor, but the specific mechanisms are still unclear. FOXF2's function and abnormal regulation in cancers are summarized in Table 1.

### *FOXF2 as a tumor promoter*

The few studies on FOXF2 as a cancer promoter have only looked at breast cancer, non-small cell lung cancer (NSCLC), melanoma, and rhabdomyosarcoma. Among these, NSCLC and breast cancer are the two most common cancers with controversial effects of FOXF2.

Breast cancer is the most commonly diagnosed cancer and the leading cause of cancer death among females worldwide, especially in developed countries (*Torre et al., 2015*).

**Table 1  The function and abnormal regulation of FOXF2 in multiple cancers.**

| Cancers | Dysregulation | Roles |
| --- | --- | --- |
| Esophagus | down regulation | FOXF2 deficiency correlates with lymph node metastasis and poor prognosis of ESCC patients[1]. |
| Gastric | down regulation | FOXF2 deficiency inhibits $\beta$-catenin degradation and promotes Wnt/$\beta$-catenin signaling pathway inducing GC growth and proliferation[2]. |
| Hepatic | down regulation | FOXF2 deficiency promotes the metastasis of HCC via inducing mesenchymal-epithelial transition[3]. |
| Colorectal | down regulation | MiR-182 down regulates FOXF2 expression to promote cell growth and metastasis through increasing $\beta$-catenin expression. |
| Prostate | down regulation | MiR-182-5p inhibits FOXF2 expression facilitating prostate cancer cells growth, proliferation and invasion. |
| Ovarian | down regulation | FOXF2 expression is down- regulated by miR-182 indicating cell growth, proliferation and metastasis. |
| Cervical | down regulation | FOXF2 deficiency promotes Wnt/$\beta$-catenin signaling pathway and EMT. |
| Breast | down regulation | (1) Down regulation of FOXF2 promotes the early-onset metastasis and poor prognosis for patients with histological grade II and TNBC[4]. (2) Deletion of FOXF2 promotes the metastasis of BLBC because FOXF2 can inhibit EMT[5]. (3) MiR-182 acts on FOXF2 to promote cell proliferation and migration of TNBC. (4) Myc-associated zinc finger protein is regulated by FOXF2 to promote proliferation and inhibit migration of BLBC. (5) FOXF2 is down regulated in luminal and HER2+ breast cancer[6]. [Lo's team] |
| Breast | up regulation | FOXF2 is highly expressed in BLBC as an EMT promoter. [Lo's team] |
| Lung | down regulation | (1) Deregulation of FOXF2 causes ECM degradation to induce EMT and cancer cells migration indicating poor prognosis in non-small cell lung cancer patients[7]. (2) Deregulation of FOXF2 promotes EMT in small cell lung patients. |
| Lung | up regulation | In non-small cell lung cancer, FOXF2 expression is up regulated and is positively associated with EMT-TFs inducing cell metastasis and invasion via inhibiting epithelial markers and miR-200 family[8]. |

**Notes.**
[1] ESCC, esophageal squamous cell carcinoma.
[2] GC, gastric cell.
[3] HCC, hepatic cellular cancer, EMT, epithelial-mesenchymal transition.
[4] TNBC, triple-negative breast cancer.
[5] BLBC, basal like breast cancer.
[6] HER2, human epidermal growth factor receptor-2.
[7] ECM, extracellular matrix.
[8] EMT-TFs, Epithelial-Mesenchymal Transition Transcription Factors.

The top two aggressive types of breast cancer are BLBC and triple-negative breast cancer (TNBC), which have similar tissue morphology, immunophenotypes, and gene expression profiles (*Manica et al., 2017*). FOXF2 is specifically expressed in TNBC/BLBC and has lower expression in luminal and HER2+ breast cancers (*Lo et al., 2016*). Lo's (*Lo et al., 2016*) results suggested that FOXF2 promoted EMT, while and Feng's research (*Yu et*
*al., 2017*) indicated that FOXF2 is a cell proliferation promoter, mainly in TNBC. First, FOXF2 interacts with EMT-TFs and TGF signaling. FOXF2 and TGF$\beta$ signaling form positive feedback and FOXF2 expression is elevated in TGF$\beta$-induced EMT. Increased FOXF2 inhibits epithelial marker expression and disrupts epithelial polarity and cell junctions by activating EMT-TFs expression (*Meyer-Schaller et al., 2018*). Second, FOXF2 reprograms tumor cells into bone-like cells by activating BMP4 /SMAD1 signaling that mediate epithelial-bone metastasis (EOM) (*Wang et al., 2019b*). By interacting with the microenvironment, breast cancer cells have the potential for bone metastasis and osteolytic destruction (*Wang et al., 2019b*). Third, FOXF2 can be activated by SP1 and myc-associated zinc finger protein (MAZ) by binding to its GC region, which inhibits the expression of mesenchymal markers (fibronectin 1 and $\alpha$-SMA) and EMT-TFs (TWIST1 and SNAIL2) and promotes BLBC proliferation (*Black, Black & Azizkhan-Clifford, 2001*; *Tian et al., 2015*; *Yu et al., 2017*).

In NSCLC, FOXF2 has two opposite effects. In this section, we will discuss its cancer development mechanism. As a tumor promoter, FOXF2 is overexpressed and positively correlated with EMT-TFs, including ZEB1, Snail, and Twist, in mesenchymal-like metastatic cells. FOXF2 induced EMT by promoting expression of the mesenchymal marker vimentin and inhibiting expression of epithelial markers E-cadherin and miR-200, thereby accelerating metastasis and lung cancer cell invasion (*Kundu et al., 2016*). MiR-200, acting as a tumor suppressor, forms two negative feedback loops with FOXF2 and ZEB1, respectively, which inhibit EMT and play a consistent role in breast cancer development (*Meyer-Schaller et al., 2018*). Moreover, FOXF2 regulates lncRNA H19/phosphate and the tensin homolog deleted on chromosome ten (PTEN) axis to accelerate the proliferative and migratory abilities of NSCLC cells in vitro (*Xu et al., 2019b*).

In melanoma, FOXF2 increases the expression of aryl hydrocarbon receptor nuclear translocator (ARNT)-related genes that form a complex with aryl hydrocarbon receptor (AHR) (*Leick et al., 2019*). The AHR: ARNT complex participates in the upregulation of the epidermal differentiation complex (EDC) gene of keratinocytes, thereby promoting melanoma development (*Furue et al., 2017*). High ARNT expression is associated with shorter overall survival (OS), and increased FOXF2 may serve as a sign of poor prognosis in melanoma (*Leick et al., 2019*). In rhabdomyosarcoma, FOXF2 inhibits the transcription of cyclin-dependent-kinase (CDK) inhibitor p21Cip1 and increases cell cycle regulatory proteins (CDK2, CDK4/6, cyclin D1, cyclin E, and cyclin A2) to promote G1-S transition and cell proliferation by activating the CDK2-RB-E2F signaling pathway (*Milewski et al., 2017*). FOXF2's regulatory mechanism as a pro-cancer factor is shown in Fig. 1.

### FOXF2 as a tumor suppressor

FOXF2 is down-regulated in gastric cancer, liver cancer, colorectal cancer, ovarian cancer, prostate cancer, small cell lung cancer, oral carcinoma (*Wang et al., 2020*), ambiguous non-small cell lung cancer, and breast cancer.

As a tumor suppressor, FOXF2 inhibits Wnt/$\beta$-catenin and CDK2-RB-E2F signaling to reduce cell growth and induce apoptosis. In gastric cancer, Akira discovered a new signal cascade and found that FOXF2 interfered with Wnt/$\beta$-catenin signaling target

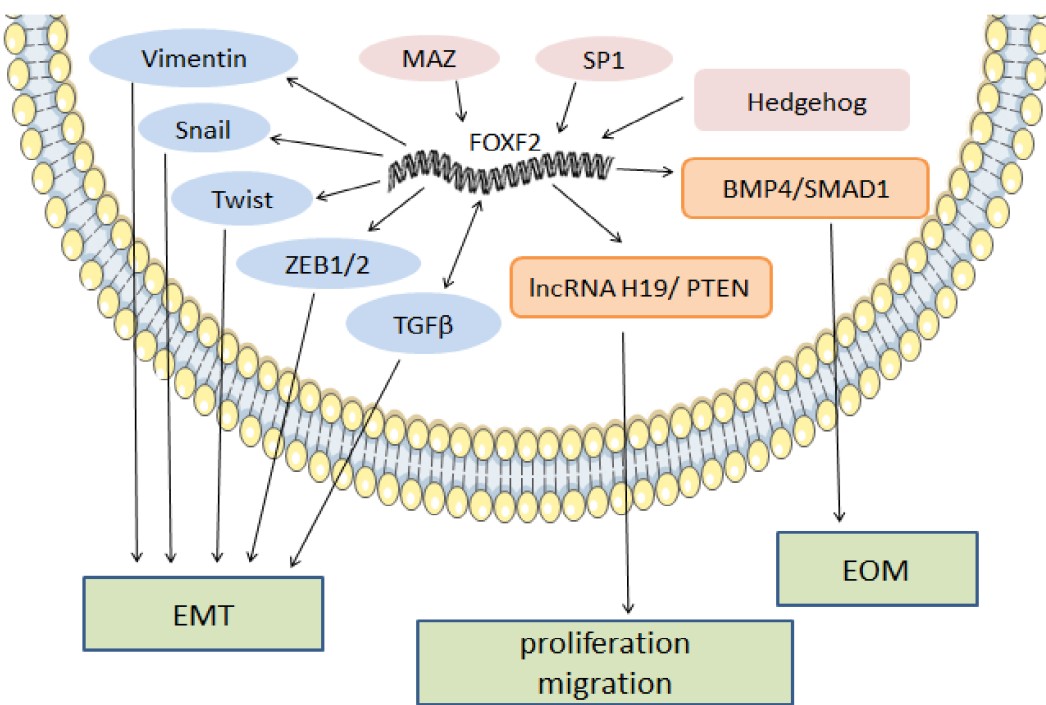

**Figure 1 The regulatory mechanism of FOXF2 as a pro-cancer factor.** MAZ, SP1 and Hh pathway-induced FOXF2 expression promote proliferation of BLBC. FOXF2 transcriptionally activates EMT-TFs (Snail, Twist, and Zeb1/2) and mesenchymal markers (Vimentin) to promote EMT in NSCLC. FOXF2 can stimulate the activation of related bone genes and reprogram cancer cells into a bone-like cell mediating EOM by activating the BMP4 / SMAD1 signaling pathway.

gene expression by directly binding to E3 ubiquitin ligases interferon regulatory factor 2-binding protein-like (IRF2BPL) that induced $\beta$-catenin ubiquitination and degradation (*Higashimori et al., 2018*). Lgr5(+) is a molecular marker of colorectal cancer (de Sousa e (*de Sousa e Melo et al., 2017*) that is associated with gastric cancer development (*Leushacke et al., 2017*). FOXF2 reduces Lgr5(+) stem cell expression by up-regulating the expression of secreted frizzled related protein 1 (SFRP1), an inhibitor of Wnt (*Nik et al., 2013*), which reduces the excessive proliferation of epithelial cells in order to maintain intestinal homeostasis (*Van den Brink & Rubin, 2013*). In cervical cancer Hela cell lines, FOXF2 reduced the expression of nuclear $\beta$-catenin and its target genes including c-myc, cyclinD1, matrix metalloproteinase 9 (MMP9), and Lgr5 (*Zhang et al., 2018*). In breast cancer, Lo suggested that FOXF2 inhibited excessive DNA replication and induced G1 arrest by blocking CDK2-RB-E2F cascade signaling (*Lo et al., 2016*).

FOXF2 inhibits EMT and MET (a reversed process of EMT) in order to prevent cell depolarization and metastasis. In hepatic carcinoma cancer, FOXF2 deficiency promotes MET by increasing epithelial marker expression and decreasing mesenchymal marker expression, which increases the colonization ability of new sites after tumor metastasis (*Dou et al., 2017*). In breast cancer, Feng's team (*Wang et al., 2015*) found that FOXF2 was an EMT inhibitor due to several factors. First, FOXF2 inhibited EMT development by directly

targeting EMT-TFs, including TWIST1, ZEB1, ZEB2, and Snail, which negatively regulates their expression (*Wang et al., 2015*). High vimentin expression mediated by EMT-TFs in lung and liver metastasis indicated that decreased FOXF2 expression made cancer cells prone to visceral metastasis (*Wang et al., 2015*). Second, FOXF2 indirectly interrupted EMT through co-expression with other transcription factors. The tumor-promoters FOXC2 and FOXQ1 are positively correlated with malignant progression and poor prognosis (*Borretzen et al., 2019*; *Kang et al., 2019*). In BLBC, FOXF2 binds to the FOXC2 promoter in order to repress FOXC2-regulated EMT and drug resistance (*Cai et al., 2015*). FOXF2 not only directly acts on target genes, but can also be co-expressed by recruiting other transcription factors. FOXF2 combined with nuclear receptor co-repressor 1 and histone deacetylase 3 jointly regulates FOXQ1 expression and exerts the EMT inhibition effect in opposition to FOXQ1 (*Kang et al., 2019*). Finally, they also confirmed that FOXF2 inhibited the transcription of the vascular endothelial growth factor-C (VEGF-C)/vascular endothelial growth factor receptor 3 (VEGFR3) signaling pathway in order to reduce lymphatic vessel formation and lymphatic metastasis inhibition in vivo and in vitro (*Wang et al., 2018b*). In small cell lung and cervical cancer, low FOXF2 expression increases mesenchymal markers, decreases epithelial markers, and induces EMT, which is conducive to cancer cell invasion and metastasis (*Chu et al., 2001*; *Oyanagi et al., 2004*; *Zhang et al., 2018*).

ECM maintenance and stabilization is one of the key factors in tumor cell metastasis prevention. In NSCLC, FOXF2 creates a stable microenvironment by synthesizing ECM, especially collagen (*Ormestad et al., 2006*). In prostate cancer, FOXF2 downregulates matrix metalloproteinase1 (MMP1) and upregulates the tissue inhibitor of metalloproteinase 3 (TIMP3) in order to reduce ECM degradation and prevent cell metastasis (*Hirata et al., 2013*). FOXF2's regulatory anti-cancer mechanism is shown in Fig. 2.

### Correlation between FOXF2 expression and prognosis

When FOXF2 acts as a tumor suppressor, its reduced expression is associated with poor prognosis. Kong showed that decreased FOXF2 expression in cancer cell lines and tissue indicated shorter disease-free survival (DFS) for patients, mainly in stage I NSCLC (*Kong et al., 2016*). In esophageal squamous cell carcinoma, decreased FOXF2 mRNA expression showed a positive correlation with lymph node metastasis, indicating a poorer OS for patients after radical resection (*Zheng et al., 2015*). Increased FOXF2 expression was associated with a good prognosis in TNBC patients (*Cai et al., 2015*). Meanwhile, decreased FOXF2 mRNA increased the risk of early recurrence and metastasis for histological grade II and TNBC patients (*Kong et al., 2013*). When FOXF2 acts as a tumor promoter, NSCLC patients with high FOXF2 levels showed a worse prognosis because FOXF2 regulated the lncRNA H19/ PTEN axis in order to accelerate the proliferative and migratory abilities of NSCLC cells in vitro (*Xu et al., 2019b*).

## Epigenetic regulation of FOXF2 in cancers
### DNA methylation of FOXF2

DNA methylation has been proven to be a carcinogenic mechanism in malignant tumors (*Morgan, Davies & Auley, 2018*). FOXF2 silenced by promoter methylation is associated with shorter survival for gastric cancer (*Higashimori et al., 2018*), breast cancer (*Tian et al.,*

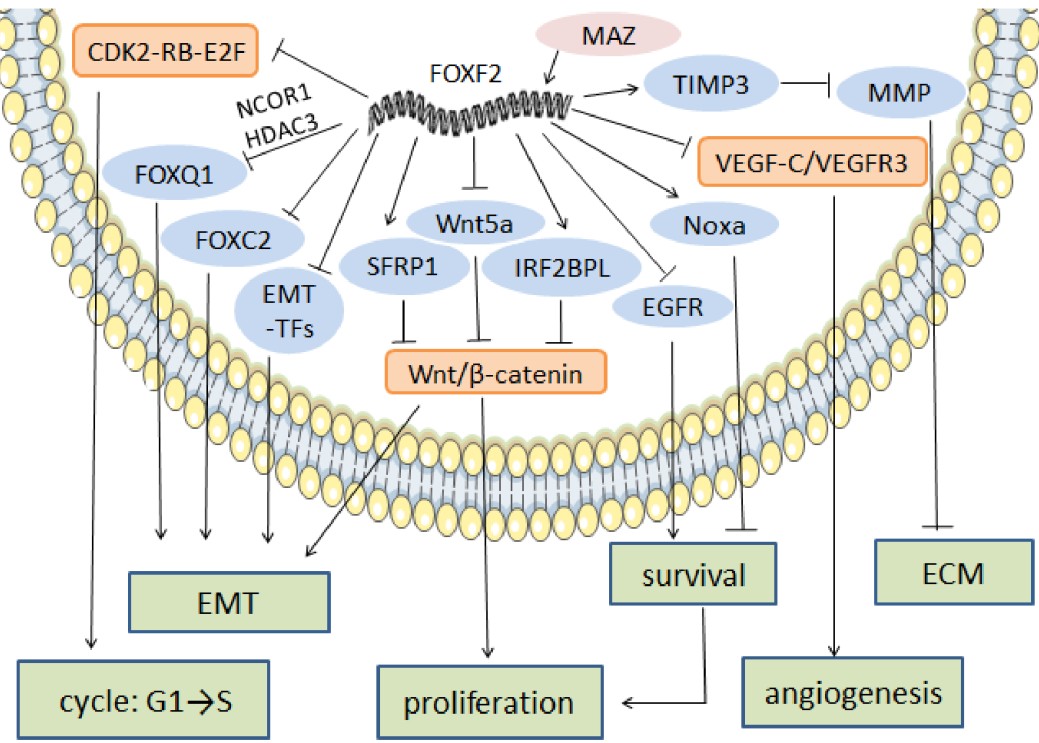

**Figure 2**  **The regulatory mechanism of FOXF2 as an anti-cancer factor.** MAZ induced-FOXF2 activation negatively regulates the mesenchymal markers and EMT-TFs in BLBC to inhibit EMT. FOXF2 inhibits the EMT properties mediated by FOXC2 and FOXQ1 that are thought to be tumor suppressors through directly binding to the FOXC2 promoter or recruiting NCOR1 and HDAC3 for co-expression, respectively. VEGF-C/VEGFR3 pathway involved in angiogenesis can also be inhibited by the expression of FOXF2. FOXF2 represses CDK2-RB-E2F, Wnt/$\beta$-catenin and EGFR pathway to inhibit excessive cell proliferation. Moreover, FOXF2 regulates ECM homeostasis by transcriptional activation of TIMP3, the inhibitor of MMP.

*2015*), and esophageal cancer patients (*Chen et al., 2017b*). DNA methylation is involved in the FOXF2 expression mechanism in different breast cancer subtypes because of low FOXF2 expression in luminal and HER2+ breast cancer due to FOXF2 methylation and high expression in non-methylated BLBC cells. Additionally, histone deacetylation is one of the silencing methods used for FOXF2 expression in luminal breast cancer cell lines (*Wang et al., 2018a*). FOXF2 methylation regulates cell proliferation by preventing SP1 from binding to the FOXF2 promoter (*Tian et al., 2015*). In gastric cancer, the Wnt/$\beta$-catenin signaling pathway is abnormally activated by FOXF2 promoter CpG island methylation (*Higashimori et al., 2018*).

### Non-coding RNA acting on FOXF2

MicroRNAs (miRNA, miR), belonging to non-coding RNAs, bind to the 3′ untranslated region (3′ UTR) of target mRNA that induce mRNA cleavage or repress the translation from mRNA to protein, which regulate target gene expression. In NSCLC, miR183-96∼182 inhibits ZEB1 transcription by regulating FOXF2 mRNA expression, thereby inhibiting

EMT. MiR183-96~182 is downregulated and ZEB1 is highly expressed in metastatic mesenchymal cells lines (*Kundu et al., 2016*). MiR-182 and miR-130a are highly expressed in colorectal cancer and target FOXF2 which acts as a tumor promoter that upregulates β-catenin and induces tumor cell growth, proliferation, and metastasis (*Chen, Tong & Yu, 2017a*; *Zhang et al., 2015*). Increasing FOXF2 expression by downregulating miR-519a restores FOXF2's tumor suppressive function and prolongs the survival of hepatic carcinoma patients (*Shao et al., 2015*). MiR-301 not only affects the PTEN-PI3K/Akt signaling pathway in order to promote angiogenesis, but it also activates the wingless-type MMTV integration site family member 5a (Wnt5a) in Wnt signaling that can inhibit FOXF2 when promoting cell proliferation (*Shi et al., 2011*). In ovarian and rectal cancer (*Wang, Yuan & Chen, 2019a*), lncRNA ADAMTS9-AS2, an endogenous competing RNA of miR-182-5p, normalizes FOXF2, which suppresses tumor metastasis by inhibiting the pro-cancer effect of miR-182-5p (*Wang et al., 2018a*). The MiR-96-5p /FOXF2 axis enhances the expression of CDK4, cyclin D1, matrix metalloproteinase 2 (MMP2), and MMP9 to promote the proliferation, invasion, and EMT of oral squamous cell carcinoma (*Wang et al., 2020*). MiR-19a-3p mediates FOXF2/Wnt signaling and increases P-GSK-3, β-catenin, N-cadherin, and Vimentin levels when promoting colorectal cancer (*Yu et al., 2020*).

## The regulatory pathway associated with FOXF2
### The hedgehog pathway

The hedgehog (Hh) signaling pathway first discovered in common drosophilae, primarily controls embryonic development, cell proliferation, tissue differentiation, and maintenance of the somatic stem cells and pluripotent cells that are essential for tissue repair (*Skoda et al., 2018*). Binding between Hh ligands and transmembrane Patched (PTCH) receptors on the cell membrane activates the Hh signaling pathway, which triggers removal of PTCH suppression on SMO. A protein complex containing kinesin protein (KIF7) and suppressor of fused (SUFU) bound to GLI zinc-finger transcription factors is dynamically transported to the activated SMO. SMO acts as a positive regulator of the Hh signaling pathway when further promoting the separation of SUFU and GLI in the cytoplasm. GLI then translocates into the nucleus and binds transcriptional targets to regulate cellular gene expression, such as cyclin D1, cyclin D2, Bcl2, and FOXF2 (*Iqbal et al., 2016*). The Hh pathway carries out the potential regulation of FOXF2 expression in Fig. 3. Aberrant activation of the Hh signaling pathway caused by mutations in related genes or Hh signaling molecule overexpression can lead to developmental abnormalities and diseases (*Wu et al., 2017*). Gorlin syndrome (*Smith et al., 2014*), basal cell carcinoma (*Gutzmer & Solomon, 2019*), medulloblastoma (*Tamayo-Orrego & Charron, 2019*), and digestive and respiratory tumors all involve Hh signaling (*Wang et al., 2013*).

Shh, the Hh ligand secreted from the pulmonary epithelium, is an important paracrine signal between the epithelium and the pulmonary vascular network, and it can promote VEGF and angiogenin production (*Zavala et al., 2017*). Shh signaling is essential for craniofacial development (*Chiang et al., 1996*). FOXF2 regulates palatal shelf growth in the Shh-FOXF2- fibroblast growth factor 18 (FGF18)-Shh molecular network. FOXF2 is

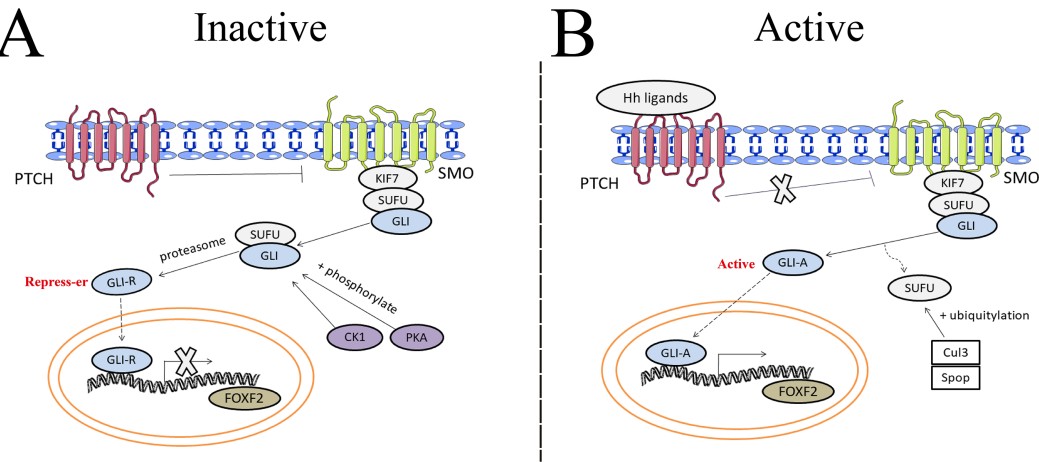

**Figure 3** **The potential regulation of FOXF2 expression by the Hedgehog pathway.** (A) Inactive signaling: In the absence of Hh ligands, PTCH precludes SMO localization to the cytomembrane and represses SMO activity. As a result, GLI cannot be separated from SUFU in the cytoplasm. Subsequently, GLI is processed into transcriptional repressors (GLI-R) by PKA- or CK1-dependent phosphorylation and proteasome. GLI-R, which has been translocated into the nucleus, cannot perform transcriptional functions. (B) Active signaling: once the Hh ligands bind to PTCH receptors on the cell membrane, the inhibitory effect of PTCH on SMO is abrogated. A protein complex containing KIF7 and SUFU bound to GLI zinc-finger transcription factors is dynamically transported to the activated SMO. Active SMO promotes release of GLI proteins from SUFU, resulting in nuclear accumulation of active GLI (GLI-A) and activation of FOXF2, the transcriptional target of Hh-GLI pathway. SUFU is degraded by Cullin-3 (Cul3) - or speckle type BTB/POZ protein (Spop) -dependent ubiquitylation.

activated by the Shh signal and in turn, inhibits FGF18 expression in order to maintain normal Shh signal expression in the epithelium (*Xu et al., 2016*). Shh-FOXF2 promotes the proliferation of cranial neural crest cell (CNCC) mesenchyme during upper lip morphogenesis, and CNCC is required for upper lip closure. Shh or FOXF2 defects reduces proliferation in the medial process mesenchyme, which is the pathological mechanism that causes cleft lips (*Everson et al., 2017*). Shh and BMP4 signals are activated in a complementary pattern in mouse embryonic mandibular arches (*Xu et al., 2019a*). BMP4 activation induced by abnormal Shh is used to maintain the physiological function of the mandible, but Shh causes FOXF2 expression downregulation that causes abnormal tongue development (*Millington et al., 2017*).

### The Wnt/β-catenin pathway

Wnt/β-catenin is an evolutionarily conserved signaling pathway that plays an important role in embryonic development and central nervous system formation by activating the transcriptional activity of target genes through β-catenin nuclear translocation. This regulates cell growth, proliferation, differentiation, and migration (*Clevers & Nusse, 2012*). Wnt/FOXF2 promotes the early differentiation of central endothelial cells and induces the expression of BBB markers ABCB1, SLCO2B1, and TNFRSF19 (*Hupe et al., 2017*; *Reyahi et al., 2015*). In addition to its role in BBB maturation, Wnt is also involved in gastrointestinal

tract development. As a key molecule in the Wnt/$\beta$-catenin signaling pathway, wingless-type MMTV integration site family member 3a (Wnt3a) stimulates intestinal stem cell markers such as Msi1, Ephb2, Dcamkl1 and distal gut-associated mesoderm gene FOXF2 (*Cao et al., 2011*). Besides, FOXF2 reduces $\beta$-catenin expression and activates BMP4 expression through the FOXF2/Wnt/BMP4 axis, which limits excessive intestinal epithelial cell proliferation in order to maintain normal intestinal homeostasis (*Ormestad et al., 2006*). When *FOXF2* was mutated, decreased BMP4 and increased Wnt5a expression were observed in fibroblasts (*Nik et al., 2013*). Therefore, FOXF2 can be regulated by the Wnt pathway and can also target the Wnt pathway when regulating downstream genes. Additionally, the abnormal FOXF2/Wnt signaling pathway enhances the stemness of Lgr5(+) cells, which increase the incidence of adenomas and susceptibility to malignant tumors (*Nik et al., 2013*).

### The TGF$\beta$-SMAD pathway

During normal cell cycle regulation, the TGF$\beta$-SMAD pathway can inhibit over-proliferation by downregulating cyclin, CDK, and c-myc expression, and by blocking cells in the G1 phase. FOXF2 is both the upstream medium and the target of the pathway. During BBB formation, FOXF2 mediates the proliferation and differentiation of pericytes and endothelial cells through the TGF$\beta$-SMAD pathway (*Reyahi et al., 2015*). In breast epithelium, TGF$\beta$ induces FOXF2 to increase caspase-dependent programmed cell death, increasing the levels of cleaved caspase-3 and its downstream cleavage target poly-(ADP-ribose) polymerase (PARP) (*Meyer-Schaller et al., 2018*). In addition, FOXF2 induced by TGF$\beta$ inhibits EGFR-PI3K/AKT-mTOR survival signaling by preventing EGFR-ligand (betacellulin and amphiregulin) activation and repressing the expression of Id2, which is a proliferation promoter (*Meyer-Schaller et al., 2018*).

During tumor development, the abnormal TGF$\beta$-SMAD signaling pathway does not exert a growth inhibitory effect, but may interact with other pathways to accelerate tumor microenvironment formation and mediate the reaction between tumor cells and ECM (*Narayan, Thangasamy & Balusu, 2005*). Since the different FOXF2 mechanisms are not completely known, there may also be differences in the effects of the related TGF$\beta$-SMAD pathway. On one hand, hypoxia enhances TGF$\beta$1 expression and SMAD2 phosphorylation when promoting EMT in lung adenocarcinoma (*Geng et al., 2016*). In TGF $\beta$ induced EMT, FOXF2 upregulation is essential for the transcriptional inhibition of E-cadherin, destruction of cell adhesion, and EMT-related cell migration. On the other hand, TGF $\beta$ and FOXF2 form a negative feedback loop mediated by miR-182-5p and SMAD3 in BLBC (*Lu et al., 2020*). Decreased FOXF2 expression promotes TGF$\beta$ signaling, which conversely silences FOXF2 by upregulating miR-182-5p. FOXF2 deficiency facilitates the visceral metastasis of BLBC (*Lu et al., 2020*).

## CONCLUSIONS

FOXF2 is a transcription factor that is specifically expressed in mesenchyme and maintains homeostasis by regulating epithelium-mesenchyme interactions and the biological behavior of cells. FOXF2 is essential for normal craniofacial, nervous, respiratory, digestive, and

genitourinary system development during the embryonic and adult stages. However, abnormal FOXF2 expression occurs in a variety of human diseases, such as Axenfeld-Rieger syndrome, stroke, osteoporosis, IUA, lung cancer, gastric cancer, and breast cancer. *FOXF2* gene mutation and abnormal activation of related regulatory signaling pathways lead to uncontrolled FOXF2 expression. Epigenetic modifications that regulate FOXF2 expression, including DNA methylation and histone acetylation, are also involved in the occurrence of cancer.

Curiously, the role of FOXF2 in cancer is not fully understood. FOXF2 is often used as a tumor suppressor in gastrointestinal and reproductive tumors, and decreased FOXF2 expression indicates cell growth and metastasis that promote cancer progression. However, FOXF2 shows two opposite effects in breast cancer and lung cancer, serving as either a tumor-promoter or a tumor-suppressor by regulating cell cycles and metastasis. While the exact mechanism for FOXF2's different roles in cancer is unclear, the following may be part of the explanation. First, FOXF2 SNPs have been found to affect not only stroke, but also cancer. SNP rs1711973 increased the risk of bladder cancer susceptibility in people who have never smoked (*Figueroa et al., 2014*). SNP rs1711972 improved the OS and DFS for NSCLC patients after curative surgery (*Seok et al., 2017*). Second, the interaction between FOXF2 and the tumor microenvironment cannot be ignored. Previous studies experiments artificially regulated gene and protein expression in vitro, which meant adding plasmids or knocking down FOXF2. There may be different effects caused by interactions with the internal environment after in vivo application. Third, FOXF2 activates different EMT-TFs that are each co-expressed with different factors in different types of tumors. Since FOXF2 can directly or indirectly recruit other transcriptional regulators, its role in regulation can be verified using knockout tests. Feng's team and Lo's team may have had different conclusions about FOXF2 expression in TNBC/BLBC because they used different experimental models and methods.

FOXF2's various functions suggest that it may be an important target for cancer treatment. Interfering with upstream pathways and using small RNAs to target FOXF2 can be potential mechanisms used in future drug production. EMT-TFs are indispensable contributors to malignant tumor metastasis, and different EMT-TFs expressed in specific organs or cell types are also specific to their target genes, to some extent (*Stemmler et al., 2019*). Therefore, disrupting EMT-TF activation using a transcription factor with a synergistic effect and FOXF2 may help prevent tumor metastasis. FOXF2/VEGF-C/VEGFR3 and FOXF2/BMP/SMAD axes are potential therapies that could inhibit the metastasis of breast cancer cells to blood vessels and bone. MiR183-96~182 and lncRNA ADAMTS9-AS2 promoters, which block EMT by increasing FOXF2 expression, can be used as treatments for lung and ovarian cancer.

Therefore, future research should explore whether SNPs in FOXF2 could be a potential predictor of the occurrence and progression of common diseases. Additionally, more small RNAs targeting FOXF2 in common diseases are waiting to be discovered, and the function of FOXF2 in lung cancer and breast cancer is still controversial. It is necessary to explore these specific mechanisms. Does FOXF2 regulate other cellular behaviors that control cancer progression? After all, cancer's various mechanisms remain unclear and

complicated. A better elucidation of FOXF2's regulatory mechanism and how it inhibits or maintains cancer by regulating gene transcription may make it an attractive target for future anti-tumor intervention.

### Funding

This study was supported by the 'Cuiying Scientific and Technological Innovation Program' of Lanzhou University Second Hospital (CY2018-MS10), the Science and Technology Plan Project of Gansu (18YF1FA108), and the National Natural Science Foundation of China (Grant No.81560343). The funders had no role in study design, data collection and analysis, decision to publish, or preparation of the manuscript.

### Grant Disclosures

The following grant information was disclosed by the authors:
Lanzhou University Second Hospital: CY2018-MS10.
Science and Technology Plan Project of Gansu: 18YF1FA108.
National Natural Science Foundation of China:  81560343.

### Competing Interests

The authors declare there are no competing interests.

### Author Contributions

- Qiong Wu conceived and designed the experiments, performed the experiments, analyzed the data, prepared figures and/or tables, authored or reviewed drafts of the paper, and approved the final draft.
- Wei Li analyzed the data, prepared figures and/or tables, authored or reviewed drafts of the paper, and approved the final draft.
- Chongge You performed the experiments, prepared figures and/or tables, authored or reviewed drafts of the paper, and approved the final draft.

### Data Availability

   This article does not use raw data; this is a literature review.

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
