# Peer review of "The regulatory roles and mechanisms of the transcription factor FOXF2 in human diseases"

_PeerJ, doi:10.7717/peerj.10845_

## Round 0.1 · original submission · Major Revisions

As noted by our reviewers, the current manuscript lacks novelty, especially compared to He et al., Cell death & disease, 2020. For acceptance of your manuscript, I would like you to focus on modifying the manuscript to bring in novelty. Also, I would strongly suggest you to get the manuscript assessed by a native English speaker for grammar and syntax. Additionally, as both the reviewers have suggested, include a schematic.

Reviewer 1 ·

Basic reporting

1. References are missing in some instances. For eg. Line 89-90 "FOXA1......identity.".

2. Although the introduction is mostly descriptive , the information listed at times is tangential and does not provide a direct connection. For eg., Line 96-99 " In the early..............is not expressed. Here authors have used words like major role and greater function rather than specifically mentioning the "role" and "function".

3. Line 129 Abbrevivated form 'EMT' is used without the description. Such an instance is repeated multiple times.

4. Line 131, The word "active" in that sentence does not seem right. Authors might want to use "activate" instead.

5. Line 159- 'FOXF2 was downregulated' should be presented in present tense. Such an instance is repeated multiple times and is almost present in the entire text.

6. Line 261 "The specific statement has been mentioned in the article of Lo (Lo 2017)." Which specific statement is author talking about. Authors should present their view on such findings rather talking about interpretations from a primary literature.

7. Line 280-281 "As mentioned above, FOXF2 is a tumor suppressor in cervical cancer." Not clear which part of the review is author refering to.

8. A schematic of FOXF2-mediated signalling in different cancers would be helpful for the readers.

Experimental design

Refernces are missing in some statements. Authors should carefully verify that refernces are appropriately placed.

Validity of the findings

No comment

·

Basic reporting

The manuscript “FOXF2: A key regulator in health and cancer” by Wu and You describes the role of FOXF2 in the development of various organs during embryonic development. The abnormal expression of FOXF2 results in dysregulation of various signaling pathways which further leads to the progression of various types of cancers. Authors have first described the importance of FOX transcriptional factor family and then detailed the effect of aberrant FOXF2 expression in relation to cancer in different organs. The mechanism of cancer progression in lungs, digestive tract, breast cancer, cervical and prostate cancer associated with FOXF2 expression has been explained. The pathways affected by FOXF2 and their effect on cancer progression are described in detail. The manuscript is concluded with author’s view to target FOXF2 for the treatment of different cancers. Overall, this review is very informative and provide the complete information of FOXF2 as a regulator in cancer. This is a well written manuscript.

Unfortunately, the study is not new. A similar review has been published on the role of FOXF2 by He et al. in the journal “Cell Death and Disease” on June 5, 2020. They have described each and every aspect of FOXF2 in embryonic development as well as in various cancers. The signaling pathways through which FOXF2 might act, have also been explained very well. The current manuscript looks very similar to this one. How do authors explain this? What is new in current manuscript in addition to this above-mentioned review article? Authors should revise the manuscript so that it does not look just a repetition of already published review.

Authors should make at least 1 schematic figure to show the role on FOXF2 in various cancers and the signaling pathways involved in which it can be indicated where the upregulation or downregulation of FOXF2 affect a particular pathway that leads to a specific type of cancer.

Manuscript should be checked thoroughly for English word choice, grammar and sentence organization. Several sentences seem to be repetitive throughout the manuscript. Also, at multiple places, the text seems discontinuous.

Authors have mentioned that FOXF2 can be a good target for the treatment of cancer, though the strategy to do so is not described. FOXF2 regulation is highly important, so its reduced or enhanced expression will not be a cure for cancer. What do authors think about a possible mechanism which can be used to target FOXF2 for cancer therapeutics?

Experimental design

no comment

Validity of the findings

no comment

---

## Round 0.2 · Minor Revisions

Dr. You,

Please address the concerns raised by our reviewer before I could accept your manuscript. I would would strongly suggest you to get the manuscript checked by a professional service for English language.

·

Basic reporting

The revised submission of the manuscript entitled “The role and regulatory mechanism of FOXF2 in diseases”, features the regulatory mechanism of FOXF2 in cancer as well as non-cancerous diseases. Authors have answered all the comments/concerns point-wise and improved the manuscript accordingly. In the revised version of the manuscript, authors have added three schematic figures for better understanding of the mechanism.

English language still needs improvement. At multiple places, it is hard to understand the sentences.

Authors have used the word “non-cancer diseases” multiple times in the text. It should be rephrased.

Title should be modified to be more specific. Disease is a very broad term. Please do specify the title.

Line 44: “It has been demonstrated that abnormal transcription factor regulation affects the expression modification of their regulatory genes, which is a mechanism required for the formation of human malignant tumors.” This line is not clear. Please do modify it.


Figure 1 and 2 look good. However, figure 3 is bit confusing. How SUFU inhibits GLI outside nucleus and afterwards figure displays GLI and FOXF2 in nucleus which is not clear. Please do modify the figure accordingly so that it describes everything clearly.

Experimental design

no comments

Validity of the findings

no comments

---

## Round 0.3 · accepted · Accept

Dear Dr. You,

Congratulations, your manuscript has been accepted for publication.

Reviewer 1 ·

Basic reporting

N/A

Experimental design

N/A

Validity of the findings

N/A

Additional comments

Congrats

·

Basic reporting

No comments

Experimental design

No comments

Validity of the findings

No comments